# Family satisfaction with critical care in the UK: a multicentre cohort study

Paloma Ferrando,[1] Doug W Gould,[1] Emma Walmsley,[1] Alvin Richards-Belle,[1] Ruth Canter,[1] Steven Saunders,[1] David A Harrison,[1] Sheila Harvey,[2] Daren K Heyland,[3,4] Lisa Hinton,[5] Elaine McColl,[6] Annette Richardson,[7] Michael Richardson,[8] Stephen E Wright,[7] Kathryn M Rowan[1]

For numbered affiliations see end of article.

**Correspondence to**
Professor Kathryn M Rowan;
kathy.rowan@icnarc.org

## ABSTRACT

**Objective** To assess family satisfaction with intensive care units (ICUs) in the UK using the Family Satisfaction in the Intensive Care Unit 24-item (FS-ICU-24) questionnaire, and to investigate how characteristics of patients and their family members impact on family satisfaction.

**Design** Prospective cohort study nested within a national clinical audit database.

**Setting** Stratified, random sample of 20 adult general ICUs participating in the Intensive Care National Audit & Research Centre Case Mix Programme.

**Participants** Family members of patients staying at least 24 hours in ICU were recruited between May 2013 and June 2014.

**Interventions** Consenting family members were sent a postal questionnaire 3 weeks after the patient died or was discharged from ICU. Up to four family members were recruited per patient.

**Main outcome measures** Family satisfaction was measured using the FS-ICU-24 questionnaire.

**Main results** A total of 12 346 family members of 6380 patients were recruited and 7173 (58%) family members of 4615 patients returned a completed questionnaire. Overall and domain-specific family satisfaction scores were high (mean overall family satisfaction 80, satisfaction with care 83, satisfaction with information 76 and satisfaction with decision-making 73 out of 100) but varied significantly across adult general ICUs studied and by whether the patient survived ICU. For family members of ICU survivors, characteristics of both the family member (age, ethnicity, relationship to patient (next-of-kin and/or lived with patient) and visit frequency) and the patient (acute severity of illness and receipt of invasive mechanical ventilation) were significant determinants of family satisfaction, whereas, for family members of ICU non-survivors, only patient characteristics (age, acute severity of illness and duration of stay) were significant.

**Conclusions** Overall family satisfaction in UK adult general ICUs was high but varied significantly. Adjustment for differences in family member/patient characteristics is important to avoid falsely identifying ICUs as statistical outliers.

**Trial registration number** ISRCTN47363549

## Strengths and limitations of this study

► This is the largest study assessing family satisfaction with intensive care unit (ICU) care.

► Unbiased selection and stratification of participating units ensured geographical spread (north, south, east and west England, Wales and Northern Ireland), hospital type (university or non-university) and ICUs of different sizes (large or small—based on number of beds) that recruited for 1 year to avoid bias from seasonal variation.

► Nesting our study within the Case Mix Programme national clinical audit was efficient and allowed for linkage of family members' to patient data.

► The same mode and timing of delivery of the Family Satisfaction in the Intensive Care Unit 24-item questionnaire were employed for family members of ICU survivors and non-survivors, avoiding potential sampling bias and allowing for meaningful comparisons between these groups.

► Despite our very large sample size, we achieved a modest response rate (58%), which was in line with previously published studies.

one of the three pillars of quality, alongside effectiveness and equity. Eliciting the views and experiences of patients is now seen as essential in delivering a high-quality service.[1] However, given that approximately 20% of patients admitted to intensive care units (ICUs) die and survivors are often unable to recall their experiences, measuring patient experience in ICU has particular challenges. For this reason, measures of family experience have been developed to help to understand the humanity of ICU care.

The most widely validated measure of family experience is the Family Satisfaction in the Intensive Care Unit questionnaire (FS-ICU).[2] This describes satisfaction, overall and in two domains—*satisfaction with care* and *satisfaction with decision-making*.[3–5] Family satisfaction reflects the extent to which perceived needs and expectations of family members are met by healthcare professionals, and

## INTRODUCTION

Humanity of healthcare, often measured as patient experience, is increasingly seen as

may be influenced by a number of factors, including families' expectations, information and communication, family-related factors (such as attitudes towards life and death, social, cultural and religious backgrounds, etc), patient-related factors (such as illness severity and whether the patient survives the ICU), hospital infrastructure and process of care.[4 6 7]

This paper reports the results of a large, prospective, multicentre, cohort study describing family satisfaction with ICU care in the UK. The overall aim of the Family-Reported Experiences Evaluation (FREE) study was to inform the potential routine use of the FS-ICU 24-item (FS-ICU-24) questionnaire for quality improvement in adult general ICUs in the UK. Specific aims were to investigate how characteristics of patients and their family members impact on family satisfaction, and to explore how family satisfaction varies across ICUs, before and after adjustment for family member and patient characteristics identified as being associated with family satisfaction.

## METHODS

This large, prospective, multicentre cohort study was nested in the Intensive Care National Audit & Research Centre (ICNARC) Case Mix Programme (CMP)—the national clinical audit of adult general ICUs in England, Wales and Northern Ireland. A stratified sample of 20 ICUs were selected to ensure geographical spread (north, south, east and west England, Wales and Northern Ireland), hospital type (university or non-university) and ICUs of different sizes (large or small—based on number of beds) that recruited for 1 year to avoid bias from seasonal variation. In accordance with care standards for UK ICUs at the time of data collection, nurse/patient ratios were 1:1 and 1:2 for level 3 (intensive care) and level 2 (high dependency) patients, respectively.

### Patient and public involvement

Engagement with patient and their family members was vital to ensuring the successful delivery of the FREE study. A former ICU patient and a family member of a former ICU patient were co-investigators on the FREE study and contributed to all aspects of the study, including design, conduct, management, analysis, interpretation of results and dissemination as members of the study management group. Additionally, the study steering committee included patient and family members.

### Recruitment and follow-up

Recruitment and follow-up of family members have been described in detail elsewhere.[8] Briefly, a 'family member' was defined as any person with close familial, social or emotional relationship to the patient and was not restricted solely to next-of-kin. Up to four family members of patients who spent >24 hours in ICU were eligible to participate if they met the following criteria: aged≥18 years, had physically visited the patient's bedside at least

once after the first 24 hours, had a UK postal address and had not already been recruited into the study.

Patients were followed-up to ICU discharge. Approximately 3 weeks after the patient had either been discharged from or died in the ICU, a questionnaire pack was mailed to their recruited and consented family member(s) direct from the ICNARC Clinical Trials Unit. Data from completed questionnaires were entered centrally onto a secure database. All identifiable information, such as names (eg, of patients, family members and ICU staff members), were removed. Quality checking of entered data was conducted and, for a 20% random sample, accuracy was verified. All fields in the database with missing data were verified against the paper questionnaires.

### Statistical analysis

Item responses were rescaled and, where relevant, reversed, according to the developer's rules, so that each response was on a scale from 0 (least satisfied) to 100 (most satisfied).[5] Recent work from our group[9] established the construct validity of the FS-ICU-24 questionnaire was improved by using three domains (splitting the *satisfaction with decision-making* domain into two—*satisfaction with information* and *satisfaction with decision-making process*). Overall family satisfaction score and three domain scores were calculated by averaging the item responses for the relevant items.

Family member and patient characteristics were described by mean and SD, median and quartiles, or number and percentage stratified by the patient outcome (alive/dead). Variation in family satisfaction was analysed across the following factors: patient, family member, ICU/hospital (hospital teaching status and number of beds in the ICU) and other contextual.

These factors were then explored using univariable and multivariable multilevel linear regression models[10] with a primary outcome of the overall family satisfaction score. Family member-level and patient-level variables that were statistically significant in the univariable models along with a priori key family member/patient variables (age and sex) were carried forward to the multivariable multilevel modelling process.[8] To reflect likely differences in the associations between factors and outcomes, separate models were fitted for family members of ICU survivors and non-survivors.

After modelling, the normality of error assumption was assessed by measurements of skewness. Normal probability plots were also used to assess the distribution of residuals at each level. As a sensitivity analysis, we ran a multilevel regression model on the square root of the score using the same set of variables to confirm inference. In secondary analyses, separate models were fitted for the three individual domains of family satisfaction. All analyses were conducted in Stata/SE V.13.0.

Variation in family satisfaction across ICUs was assessed graphically using funnel plots, which plot the average family satisfaction score for each critical care unit against the number of family members returning questionnaires.

**Table 1** Family member characteristics stratified by the patient's ICU outcome

| Family member characteristics | All family members (N=7019) | Family members of ICU survivors (N=6149) | Family members of ICU non-survivors (N=870) |
|---|---|---|---|
| Age, years, mean (SD) | 54 (15.1) | 54 (15.0) | 52 (15.2) |
| Age group, years, n (%) | | | |
| <30 | 507 (7.5) | 439 (7.4) | 68 (8.0) |
| 30–39 | 701 (10.3) | 595 (10.0) | 106 (12.5) |
| 40–49 | 1423 (21.0) | 1245 (21.0) | 178 (21.0) |
| 50–59 | 1614 (23.8) | 1406 (23.7) | 208 (24.6) |
| 60–69 | 1507 (22.2) | 1334 (22.5) | 173 (20.4) |
| 70–79 | 827 (12.2) | 747 (12.6) | 80 (9.5) |
| 80+ | 204 (3.0) | 171 (2.9) | 33 (3.9) |
| Sex, n (%) | | | |
| Male | 2327 (33.5) | 2052 (33.7) | 275 (31.9) |
| Female | 4622 (66.5) | 4034 (66.3) | 588 (68.1) |
| Ethnicity, n (%) | | | |
| White | 6555 (94.0) | 5738 (93.9) | 817 (94.6) |
| Asian | 138 (2.0) | 114 (1.9) | 24 (2.8) |
| Black | 54 (0.8) | 50 (0.8) | 4 (0.5) |
| Mixed ethnicity or other ethnic group | 88 (1.3) | 84 (1.4) | 4 (0.5) |
| Not stated | 139 (2.0) | 124 (2.0) | 15 (1.7) |
| Relationship to patient, n (%) ('I am the patient's…') | | | |
| Partner | 2096 (29.9) | 1891 (30.8) | 205 (23.6) |
| Child | 654 (9.3) | 1893 (30.8) | 346 (39.8) |
| Parent | 2239 (31.9) | 622 (10.1) | 32 (3.7) |
| Sibling | 704 (10.0) | 624 (10.1) | 80 (9.2) |
| Other relative | 969 (13.8) | 799 (13.0) | 170 (19.5) |
| Other non-relative | 356 (5.1) | 319 (5.2) | 37 (4.3) |
| Next-of-kin, n (%) | 3520 (50.2) | 3153 (51.4) | 367 (42.3) |
| Lives with patient, n (%) | 2559 (36.5) | 2311 (37.6) | 248 (28.5) |
| Highest level of education, n (%) | | | |
| NVQ level 1 or 2 | 1683 (28.9) | 1465 (28.9) | 218 (29.1) |
| NVQ level 3 | 1123 (19.3) | 989 (19.5) | 134 (17.9) |
| NVQ level 4 or 5 | 1769 (30.4) | 1537 (30.3) | 232 (31.0) |
| Other | 1244 (21.4) | 1080 (21.3) | 164 (21.9) |
| Quintile of deprivation, n (%) | | | |
| 1 (least deprived) | 1190 (17.1) | 1164 (19.9) | 159 (19.4) |
| 2 | 1405 (20.2) | 1281 (21.9) | 181 (22.1) |
| 3 | 1488 (21.4) | 1238 (21.1) | 181 (22.1) |
| 4 | 1488 (21.4) | 1189 (20.3) | 169 (20.7) |
| 5 (most deprived) | 1391 (20.0) | 989 (16.9) | 128 (15.6) |
| Distance (km) from home to hospital, median (IQR) | 12.4 (5.4–33.6) (6714) | 12 (6–34) | 12 (5–33) |
| Previous experience of ICU as a family member, n (%) | 1841 (26.6) | 1641 (27.1) | 200 (23.3) |
| Frequent visitor, n (%) | 5403 (78.9) | 4713 (78.6) | 690 (81.2) |

National Vocational Qualification (NVQ) level 1 or 2, equivalent to General Certificate of Secondary Education (GCSE) or O-level (school exams taken at age 16 years); NVQ level 3, equivalent to A-level, AS-level or High School Certificate (school exams taken at age 18 years) and NVQ level 4 or 5, equivalent to degree, higher degree, Higher National Certificate and Higher National Diploma.
ICU, intensive care unit.

Control limits placed at 2 and 3 SDs around the overall mean indicate the regions of the funnel within which we would expect that 95% and 99.8%, respectively, of points to lie if all variation was due to chance.[11]

Due to the natural structure of the data and the planned analysis multilevel multiple imputation was used to complete non-responses and partial responses for outcomes and family member characteristics. Data were imputed using REALCOM-Impute, an MLwiN V.2.15 macro that generates imputations for hierarchical data.[12] To test whether our findings were influenced by using imputed data, we also conducted sensitivity analyses using a traditional approach to scoring the FS-ICU-24 questionnaire by including only responders with ≥60% of items completed. All analyses were conducted in accordance with a predefined statistical analysis plan and reported in line with the Strengthening the Reporting of Observational Studies in Epidemiology guidance on the analysis of observational studies.[13]

## RESULTS

Of the 210 adult, general ICUs participating in the CMP, 142 (67.6%) expressed an interest in participating and the 20 ICUs were selected using stratified, random sampling. The characteristics and outcomes of all admissions to the study ICUs were similar to admissions to all ICUs in the CMP during the same period (online supplementary table S1).

Between 28 May 2013 and 30 June 2014, 18 757 patients were admitted to the 20 ICUs, of whom 12 730 patients stayed at least 24 hours in the ICU. From these, 12 346 family members of 6380 patients were recruited. Fully or partially completed questionnaires were returned by 7173 family members of 4615 patients. Family members of patients for whom no CMP data were available were not included; so, finally, 7019 were included in the final analysis (online supplementary figure S1).

Response rates varied by family member characteristics, including age, gender, ethnicity, level of deprivation (based on residential postcode), level of education and relationship with the patient. Family members documented in ICU records as next-of-kin were more likely to complete the questionnaire than those who were not, while family members for whom English was their first language were more likely to complete the questionnaire than those for whom it was not (online supplementary table S2).

A detailed description of the inclusion process, response rates and responders' characteristics has been reported in FREE study report.[8] Comparisons of family member and patient characteristics for ICU survivors and non-survivors are presented in tables 1 and 2, respectively.

Both overall and individual domain scores generally revealed high satisfaction (table 3); however, a long tail was present indicating some questionnaires were returned with very low scores (figure 1). Family members of ICU non-survivors had higher scores for overall satisfaction and satisfaction with the decision-making process domain than family members of ICU survivors.

Univariable analyses of the association between family satisfaction and characteristics, patient characteristics, ICU/hospital characteristics and contextual factors are presented in online supplementary appendix table S3–S5. There was no evidence of differences in family satisfaction according to hospital teaching status or the number of beds in the ICU; however, these variables were retained in the multilevel multivariable models due to their controlling effect on the other coefficients in the models. A summary of the candidate variables considered in the models and a justification for their inclusion/exclusion is detailed in online supplementary table S6.

Results of the multivariable multilevel models for overall family satisfaction are presented in table 4. Among family members of ICU survivors, there was evidence of an association with overall family satisfaction for family member age group, family member ethnicity, next-of-kin/lives with patient, frequency of visits, ICNARC Physiology Score and receipt of advanced respiratory support. Among family members of non-survivors, only the following patient factors were significant: patient age, ICNARC Physiology Score and ICU length of stay. These associations were significant when controlling for other predictors in the model. A priori-specified interaction terms and random slopes did not improve the fit of the models and so these terms were not retained.

Variances at both the patient and ICU/hospital levels were statistically significant but the variance partition coefficients at the ICU/hospital level were small in both the null and final multilevel models (4% and 3% for ICU survivors and 2% and 2% for ICU non-survivors, respectively), which means differences in overall family satisfaction scores were mainly at the patient and family member levels. Variance at the patient level represented 44% of the total variance in overall family satisfaction in the final models for family members of both the ICU survivors and non-survivors.

Full results of the multivariable multilevel models for the domain scores are reported in online supplementary appendix table S7–S9.

Figure 2 shows the funnel plots for the overall family satisfaction score, before and after adjustment for family member and patient characteristics from the multivariable multilevel models. Adjusting for family member and patient characteristics reduced the variability across ICUs, resulting in fewer ICUs outside the funnel plot control limits, but the relative position of ICUs remained the same. Funnel plots for the individual domain scores before and after adjustment can be found in online supplementary figure S2.

## SENSITIVITY ANALYSES

Multivariable multilevel models using the square root transformation of the satisfaction scores gave consistent results. In the models using imputed data, the direction

**Table 2** Patient characteristics stratified by ICU outcome

| Patient characteristics | All patients (N=4506) | ICU survivors (N=4007) | ICU non-survivors (N=499) |
|---|---|---|---|
| Age, years, mean (SD) | 63 (17.0) | 63 (17.3) | 68 (13.2) |
| Age group, years, n (%) | | | |
| <30 | 254 (5.6) | 246 (6.1) | 8 (1.6) |
| 30–39 | 232 (5.1) | 223 (5.6) | 9 (1.8) |
| 40–49 | 412 (9.1) | 384 (9.6) | 28 (5.6) |
| 50–59 | 643 (14.3) | 586 (14.6) | 57 (11.4) |
| 60–69 | 1100 (24.4) | 966 (24.1) | 134 (26.9) |
| 70–79 | 1159 (25.7) | 1003 (25.0) | 156 (31.3) |
| 80+ | 706 (15.7) | 599 (14.9) | 107 (21.4) |
| Sex, n (%) | | | |
| Male | 2561 (56.8) | 2264 (56.5) | 297 (59.5) |
| Female | 1945 (43.2) | 1743 (43.5) | 202 (40.5) |
| Ethnicity, n (%) | | | |
| White | 4176 (92.7) | 3706 (92.5) | 470 (94.2) |
| Asian or Asian British | 81 (1.8) | 69 (1.7) | 12 (2.4) |
| Black or Black British | 42 (0.9) | 39 (1.0) | 3 (0.6) |
| Mixed ethnicity or other ethnic group | 79 (1.8) | 74 (1.8) | 5 (1.0) |
| Not stated | 128 (2.8) | 119 (3.0) | 9 (1.8) |
| Quintile of deprivation, n (%) | | | |
| 1 (least deprived) | 774 (17.3) | 690 (17.4) | 84 (17) |
| 2 | 905 (20.3) | 812 (20.4) | 93 (18.8) |
| 3 | 928 (20.8) | 822 (20.7) | 106 (21.4) |
| 4 | 950 (21.3) | 841 (21.2) | 109 (22) |
| 5 (most deprived) | 912 (20.4) | 809 (20.4) | 103 (20.8) |
| Distance (km) from home to hospital, median (IQR) | 33.1 (67.8) 9.3 (4.3–19.9) (4475) | 10 (4–20) | 8 (4–16) |
| APACHE II severe co-morbidities, n (%) | | | |
| Liver | 124 (2.8) | 94 (2.3) | 30 (6.0) |
| Renal | 108 (2.4) | 97 (2.4) | 11 (2.2) |
| Respiratory | 146 (3.2) | 119 (3.0) | 27 (5.4) |
| Cardiovascular | 117 (2.6) | 100 (2.5) | 17 (3.4) |
| Metastatic cancer | 121 (2.7) | 110 (2.7) | 11 (2.2) |
| Haematological malignancy | 103 (2.3) | 81 (2.0) | 22 (4.4) |
| Immunocompromise | 369 (8.2) | 318 (7.9) | 51 (10.2) |
| Prior dependency, n (%) | | | |
| Able to live without assistance | 3267 (72.5) | 2944 (73.5) | 323 (64.7) |
| Minor or major assistance | 1171 (26.0) | 1004 (25.1) | 167 (33.5) |
| Total assistance | 47 (1.0) | 42 (1.0) | 5 (1.0) |
| Unknown | 21 (0.5) | 17 (0.4) | 4 (0.8) |
| Surgical status, n (%) | | | |
| Non-surgical | 2808 (62.3) | 2396 (59.8) | 412 (82.6) |
| Planned admission following elective or scheduled surgery | 702 (15.6) | 686 (17.1) | 16 (3.2) |

Continued

| Patient characteristics | All patients (N=4506) | ICU survivors (N=4007) | ICU non-survivors (N=499) |
|---|---|---|---|
| Unplanned admission following surgery of any urgency | 996 (22.1) | 925 (23.1) | 71 (14.2) |
| ICNARC Physiology Score, mean (SD) | 18 (8.3) | 18 (7.9) | 26 (8.1) |
| APACHE II score, mean (SD) | 17 (6.3) | 16 (6.1) | 21 (6.2) |
| ICU length of stay (days), median (IQR) | 4.9 (2.9–9.1) | 4.8 (2.8–9.0) | 6.0 (3.6–10.6) |
| Organ support received in the ICU, n (%) | | | |
| Advanced respiratory support | 2540 (56.4) | 2124 (53.0) | 416 (83.4) |
| Advanced cardiovascular support | 1325 (29.4) | 1037 (25.9) | 288 (57.7) |
| Renal support | 691 (15.3) | 510 (12.7) | 181 (36.3) |
| Neurological support* | 617 (13.7) | 503 (12.6) | 114 (22.8) |
| Duration (calendar days) of organ support among those receiving the support, median (IQR) | | | |
| Advanced respiratory support | 5.0 (2.0–9.0) | 4 (2–9) | 6 (4–10) |
| Advanced cardiovascular support | 3.0 (2.0–4.0) | 2 (2–4) | 3 (2–5) |
| Renal support | 4.0 (3.0–8.0) | 4 (3–8) | 4 (3–8) |
| Neurological support | 3.0 (2.0–7.0) | 3 (2–7) | 3 (2–5) |
| Death before acute hospital discharge, n (%) | 852 (19.2) | 353 (8.9) | N/A |

*Including admission receiving invasive neurological monitoring or treatment, continuous intravenous medication for seizures and/or cerebral monitoring, and therapeutic hypothermia using protocols and devices.
APACHE, Acute Physiologic Assessment and Chronic Health Evaluation; ICNARC, Intensive Care National Audit & Research Centre; ICU, intensive care unit; N/A, not applicable.

and order of magnitude of significant coefficients were similar to those estimated using the traditional approach to scoring partially completed questionnaires (online supplementary table S10 and S11). On average, the multiple imputation approach tended to identify larger numbers of potential outliers due to the larger sample sizes, and therefore narrower funnels.

## DISCUSSION

Overall and domain-specific family satisfaction measured with the FS-ICU-24 questionnaire was high. However, we found that scores vary significantly across adult general ICUs and that family members of patients who died in the ICU had higher levels of satisfaction. For family members of ICU survivors, characteristics of both the family member and the patient were significant determinants of family satisfaction, whereas, for family members of ICU non-survivors, only patient characteristics were significant. Adjustment for these family member and patient characteristics reduced the variation in family satisfaction across ICUs, resulting in fewer ICUs being identified as statistical outliers.

While the observational design of the FREE study precludes any causative inferences being made, we speculate that the higher levels of family satisfaction among family members of ICU non-survivors may be due to a number of factors, either singly or combined, including greater involvement of the family in end-of-life decision-making, family members of survivors having ongoing issues to cope with following their family member's discharge from ICU and/or other unknown factors. In order to fully identify and understand why family members of ICU non-survivors have higher family satisfaction, a detailed qualitative study is required.

The overall satisfaction score was comparable with other published studies employing similar methods to administer the FS-ICU-24 questionnaire.[14–17] Our findings are also consistent with a study by Wall et al[6] which identified that families of ICU non-survivors were more satisfied than families of ICU survivors. Similarly, Stricker et al,[7] among a number of patient-level and ICU-level factors studied, found that increasing acute severity of illness of the patient (evaluated using the Simplified Acute Physiology Score (SAPS) II score) was associated with increasing satisfaction on the overall family satisfaction

**Table 3** Overall family satisfaction score for all family members and for family members by patient outcome

| Summary measures | All family members (N=7017*) | Family members of ICU survivors (N=6147*) | Family members of ICU non-survivors (N=870) |
|---|---|---|---|
| **Overall family satisfaction score** | | | |
| Median (IQR) | 83.3 (70.4–93.0) | 82.7 (69.9–92.7) | 87.1 (74.4–94.8) |
| Mean (SD) | 79.7 (16.7) | 79.3 (16.5) | 82.0 (17.5) |
| (95% CI) | (79.2 to 80.1) | (78.9 to 79.8) | (80.9 to 83.2) |
| **Satisfaction with care domain score** | | | |
| Median (IQR) | 87.5 (74.3–96.4) | 87.5 (73.6–96.4) | 88.1 (76.8–96.4) |
| Mean (SD) | 83.1 (16.0) | 83.0 (15.9) | 83.8 (16.9) |
| (95% CI) | (82.7 to 83.4) | (82.6 to 83.4) | (82.7 to 84.9) |
| **Satisfaction with information domain score** | | | |
| Median (IQR) | 79.2 (66.7–95.8) | 79.2 (62.5–95.8) | 83.3 (70.8–100.0) |
| Mean (SD) | 76.2 (22.0) | 75.7 (22.0) | 79.6 (22.9) |
| (95% CI) | (75.7 to 76.7) | (75.1 to 76.2) | (78.1 to 81.0) |
| **Satisfaction with the decision-making process domain score** | | | |
| Median (IQR) | 75.6 (59.3–93.1) | 75.0 (57.5–88.8) | 87.5 (68.8–100.0) |
| Mean (SD) | 73.1 (22.3) | 72.1 (22.0) | 79.6 (22.9) |
| (95% CI) | (72.5 to 73.6) | (71.6 to 72.7) | (78.1 to 81.1) |

*Two family members returned questionnaires but did not complete any of the 24 FS-ICU items—responses were not imputed for these family members.

FS-ICU, Family Satisfaction in the Intensive Care Unit; ICU, intensive care unit.

score; however, lower satisfaction was associated with ICU-level characteristics of a written admission/discharge policy and a higher patient:nurse ratio.

It is of note that one of the largest magnitude associations in the FREE study was the finding that family members of white ethnicity, of both ICU survivors and non-survivors, had higher satisfaction than family

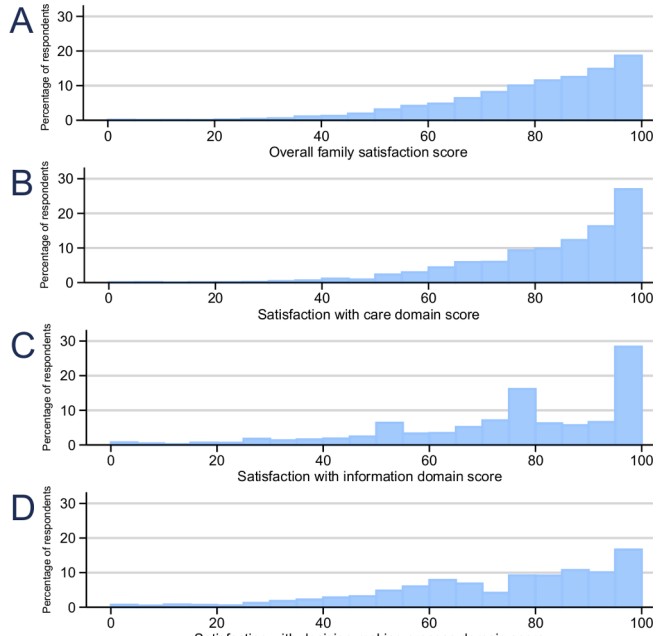

**Figure 1** Distribution of overall family satisfaction score.

members of other ethnicities. Further investigation of this issue is warranted to understand whether this reflects, for example, either cultural variation in family members' expectations or a need to engage better and communicate with family members who may not have English as their first language (17% of family members of other ethnicities indicated that their first language was not English compared with less than 1% of white ethnicity).

Our work has several important strengths. To the best of our knowledge, this is the largest study assessing family satisfaction with ICU care. Nesting our study within the national clinical audit programme was efficient and novel, and allowed for unbiased selection and stratification of participating units and linkage of family members' to patient data. One important strength is that the same mode and timing of delivery of the FS-ICU-24 questionnaire was employed for family members of ICU survivors and non-survivors, avoiding potential sampling bias and allowing for meaningful comparisons between these groups. Finally, the large sample size of family members allowed for robust multilevel multivariable modelling of factors associated with overall family satisfaction to inform important adjustment of any future assessment using this questionnaire. Despite our very large sample size, we achieved a modest response rate (58%); however, this was similar to other studies with smaller sample sizes.[6 14]

Our study does, however, have limitations. First, when assessing satisfaction, it is not uncommon for continuous measures to be skewed. While the skewed nature of the satisfaction scores does not affect the parameter estimates

**Table 4** Multivariable multilevel models for overall family satisfaction score

| Variables | Family members of ICU survivors (N=6143)* | | | Family members of ICU non-survivors (N=869)* | | |
|---|---|---|---|---|---|---|
| | Coef. | 95% CI | P value | Coef. | 95% CI | P value |
| Fixed effects—family member level | | | | | | |
| Constant | 68.30 | (63.42 to 73.17) | | 55.70 | (42.26 to 69.14) | |
| Family member age, years (vs <30) | | | 0.041 | | | 0.18 |
| 30–39 | 1.97 | (0.11 to 3.82) | | 2.01 | (−2.64 to 6.66) | |
| 40–49 | 1.65 | (0.02 to 3.29) | | 3.37 | (−1.01 to 7.75) | |
| 50–59 | 1.96 | (0.35 to 3.56) | | 4.12 | (−0.09 to 8.33) | |
| 60–69 | 1.35 | (−0.31 to 3.01) | | 4.26 | (−0.25 to 8.79) | |
| 70–79 | 1.32 | (−0.52 to 3.17) | | 5.92 | (0.69 to 11.14) | |
| 80+ | −1.34 | (−4.06 to 1.37) | | −0.18 | (−6.80 to 6.43) | |
| Family member sex—female (vs male) | 0.32 | (−0.48 to 1.12) | 0.44 | 0.66 | (−1.45 to 2.77) | 0.54 |
| Family member ethnicity—white (vs non-white) | 3.59 | (1.38 to 5.80) | 0.001 | 7.12 | (−0.00 to 14.25) | 0.050 |
| Next-of-kin/lives with patient (vs lives with patient) | | | <0.001 | | | 0.26 |
| Next-of-kin, does not live with patient | −1.39 | (−2.56 to −0.22) | | 1.08 | (−2.39 to 4.55) | |
| Not next-of-kin, does not live with patient | −2.33 | (−3.26 to −1.41) | | −1.24 | (−3.88 to 1.40) | |
| Frequent visitor | 2.83 | (1.82 to 3.84) | <0.001 | 1.53 | (−1.34 to 4.39) | 0.30 |
| Fixed effects—patient level | | | | | | |
| Patient age (per 10 years) | 0.01 | (−0.28 to 0.31) | 0.93 | 1.18 | (0.09 to 2.27) | 0.033 |
| Patient sex—female (vs male) | 0.26 | (−0.73 to 1.25) | 0.61 | 1.92 | (−0.85 to 4.70) | 0.17 |
| Dependency (vs none) | | | 0.15 | | | 0.74 |
| Minor or major | −0.30 | (−1.60 to 1.00) | | −0.22 | (−3.36 to 2.92) | |
| Total | −4.62 | (−9.32 to 0.07) | | 4.98 | (−8.10 to 18.07) | |
| Surgical status (vs non-surgical) | | | 0.63 | | | 0.82 |
| Planned elective/scheduled | −0.74 | (−2.24 to 0.77) | | −2.61 | (−10.77 to 5.54) | |
| Unplanned | −0.26 | (−1.46 to 0.94) | | −0.08 | (−3.95 to 3.80) | |
| ICNARC Physiology Score (per point) | 0.16 | (0.09 to 0.24) | <0.001 | 0.17 | (0.00 to 0.34) | 0.045 |
| ICU length of stay (per day) | −0.02 | (−0.07 to 0.03) | 0.44 | −0.30 | (−0.46 to 0.15) | <0.001 |
| Advanced respiratory support | 2.96 | (1.80 to 4.11) | <0.001 | | | |
| Fixed effects—ICU/hospital level | | | | | | |
| Hospital type (vs non-university) | | | 0.49 | | | 0.55 |
| University | 0.86 | (−3.61 to 5.32) | | −1.51 | (−7.51 to 4.50) | |
| University affiliated | 1.97 | (−1.26 to 5.20) | | 1.77 | (−2.55 to 6.09) | |
| Number of ICU beds (per bed) | −0.00 | (−0.23 to 0.23) | 0.97 | 0.26 | (−0.08 to 0.61) | 0.13 |
| Random effects—SD (SE) | | | | | | |
| Between ICUs | 2.91 | (0.60) | | 2.81 | (1.10) | |
| Within ICUs between patients | 10.94 | (0.29) | | 11.16 | (0.69) | |
| Within patients between family members | 11.98 | (0.21) | | 12.26 | (0.44) | |
| Variance partition—percentage | | | | | | |
| Between ICUs | 3% | | | 2% | | |
| Between patients | 44% | | | 44% | | |

*Five patients had missing data on age group on both the questionnaire and the web portal—due to the very small amount of missing data in this key variable, these missing values were not imputed.
Coef., coefficient; ICNARC, Intensive Care National Audit & Research Centre; ICU, intensive care unit.

A

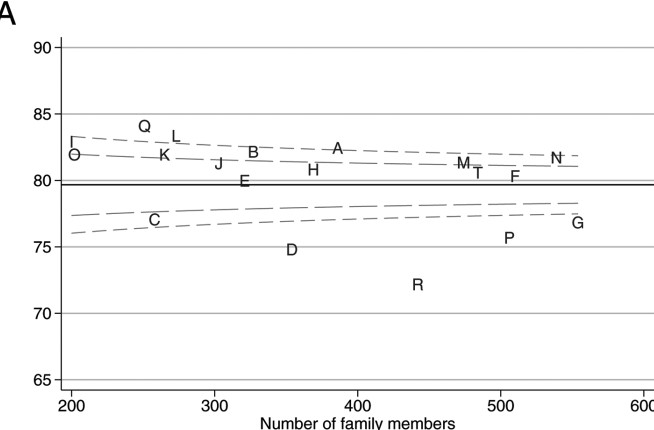

B

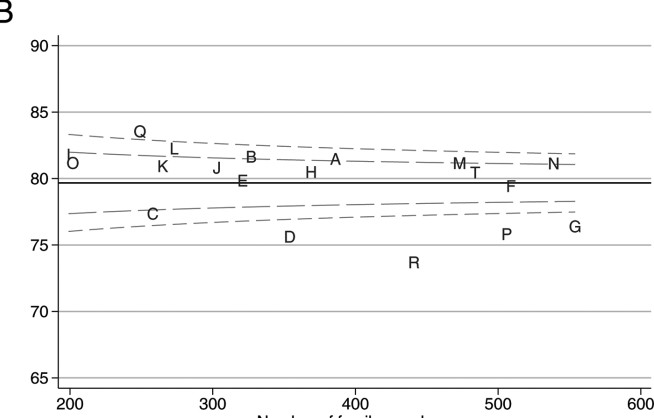

**Figure 2** Variation across ICUs in the mean overall family satisfaction score (A) before and (B) after adjustment for patient and family member characteristics. ICUs, intensive care units.

In conclusion, this large, prospective, multicentre cohort study indicated that overall family satisfaction with adult general ICU care in the UK was high. However, our findings indicate that there is scope for some UK adult general ICUs to improve. Our results suggest that the FS-ICU-24 questionnaire could be used to audit family satisfaction but adjustment for differences in family member/patient characteristics is important to avoid falsely identifying ICUs as statistical outliers.

**Author affiliations**
[1]Intensive Care National Audit and Research Centre, London, UK
[2]Global Health and Development, London School of Hygiene & Tropical Medicine, London, UK
[3]Clinical Evaluation Research Unit, Kingston General Hospital, Kingston, Ontario, Canada
[4]Department of Critical Care Medicine, Queens University, Kingston, Ontario, Canada
[5]Health Experiences Research Group, Nuffield Department of Primary Care Health Sciences, University of Oxford, Oxford, UK
[6]Institute for Health and Society, Newcastle University, Newcastle upon Tyne, UK
[7]Perioperative and Critical Care, Freeman Hospital, Newcastle upon Tyne Hospitals NHS Foundation Trust, Newcastle upon Tyne, UK
[8]Patient Representative, Newcastle, UK

**Acknowledgements** We wish to thank all the patients, family members and staff from all the units that participated in the study.

**Collaborators** Research staff at sites: C Smalley and R Jacob (Arrowe Park Hospital); S Chau, SA Pearson, K Ellis and R Watmough (Barnsley Hospital); M Faulkner, L Evans and H Robertson (Countess of Chester Hospital); P Wakefield, R Abrahams, N Summers and H Wooldridge (Darent Valley Hospital); H McMillan, S Tyson, K Tantam, S Olver, C Brown and C Tippett (Derriford Hospital); S Moreton, S Jones, A Deeney, J Gibbins and A Oglesby (Dorset County Hospital); C Randell, M Allsop, K Harris, C Scott and C Boyd (Freeman Hospital); E Coughlan, A Jefferies and K Wylie (Manchester Royal Infirmary); C Plowright, C Pegg, L Cooper and T Hatton (Medway Maritime Hospital); P Doble, P Richards, D Bayford and K Adams (Musgrove Park Hospital); J Spimpolo, M Burt and R Pillai (Northampton General Hospital); KA Simeson and S Buckley (Pinderfields Hospital); A Jackson, M Nadolski and H Baker (Royal Devon & Exeter Hospital, Wonford); N Mason, U Gunter and L Roberts (Royal Gwent Hospital); T Evans, E Cooke, M Ogden and P Dark (Salford Royal Hospital); M Cody, F Hogg and D McCahery (South West Acute Hospital); D Dawson, J Mellinghoff, S Prudden, N Poonuth and C Ryan (St George's Hospital); G Mandersloot and A Smith (The Royal London Hospital); S Hagan, L Humphries and E Murphy (Ulster Hospital); E Walker, H Payne and X Zhao (Watford General Hospital) and C Edmondson, S Anglesea and H Williams (Wrexham Maelor Hospital). Study Steering Committee: Dr Kathleen Daly (independent chair), Andrina Colquoun (independent), Dr Maureen Dalziel, Kirsty Everingham (independent), Doreen Henry (independent), Joan Pearson (independent), Catherine Plowright, Dr Laura Price (independent), Professor Kathryn Rowan, Professor Mervyn Singer (independent) and Dr Stephen Wright.

**Contributors** KMR as chief investigator conceived the idea and designed the study with DAH, SH, DKH, LH, EM, MR, AR and SEW. EW co-ordinated the study and contributed to data acquisition with AR-B, RC, SS, SH, AR and SEW. PF, DWG, DAH, SH, DKH, LH, EM, MR, SEW and KMR were involved in the analysis and interpretation of the results. All the authors were involved in drafting and editing, and have approved the final manuscript.

**Funding** This project was funded by the National Institute for Health Research (NIHR) Health Services and Delivery Research (HS&DR) Programme (11/2003/56).

**Disclaimer** The funder had no involvement in study design; collection, analysis and interpretation of data; writing of the report or decision to submit the article for publication. The views and opinions expressed herein are those of the authors and do not necessarily reflect those of the HS&DR Programme, NIHR, NHS or the Department of Health.

**Competing interests** Kathryn M Rowan is a member of the NIHR HS&DR Board. Elaine McColl was an editor for the NIHR Journals Library between 2013 and 2016 and her employers received a fee for this work. The other authors declare no conflicts of interest. All authors have completed the Unified Competing Interest form (available on request from the corresponding author).

in multilevel models,[18 19] it might cause problems when one is interested in the significance or CIs of the variance terms at higher levels.[19] In our analyses, we corrected the asymptotic standard errors using a robust (Huber/White) estimator to improve inference and performed a sensitivity analysis using a square root transformation, which did not change our conclusions. Second, by excluding family members of patients who had spent less than 24 hours on ICU—to ensure that family members had spent long enough on ICU to feel able to respond to the questionnaire—we may have missed a small group of family members of very sick patients who died soon after admission to ICU. Third, there were differences in the case mix and outcome of patients between those who had at least one family member recruited and those who did not, leading to potential bias in the results. Fourth, we found that younger family members and those from non-white ethnicities were less likely to respond and important information may have been missed. Finally, 94% of patients were of white ethnicity, which is above that of the ethnic makeup of the UK (87%) and may make the overall family satisfaction scores less generalisable to other ethnicities.

**Patient consent for publication** Not required.

**Ethics approval** The study was reviewed and approved by the National Research Ethics Service Committee South Central—Berkshire B (reference 13/SC/0037).

**Provenance and peer review** Not commissioned; externally peer reviewed.

**Data availability statement** Data are available on reasonable request.

**Open access** This is an open access article distributed in accordance with the Creative Commons Attribution 4.0 Unported (CC BY 4.0) license, which permits others to copy, redistribute, remix, transform and build upon this work for any purpose, provided the original work is properly cited, a link to the licence is given, and indication of whether changes were made. See: https://creativecommons.org/licenses/by/4.0/.

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
