## [Reviewer comments · BMJ Open]

ARTICLE DETAILS

TITLE (PROVISIONAL)	Family satisfaction with critical care in the United Kingdom: a multi-centre cohort study
AUTHORS	Ferrando, Paloma; Gould, Doug; Walmsley, Emma; Richards-Belle, Alvin; Canter, Ruth; Saunders, Steven; Harrison, David; Harvey, Sheila; Heyland, Daren; Hinton, Lisa; McColl, Elaine; Richardson, Annette; Richardson, Michael; Wright, Stephen; Rowan, Kathryn

VERSION 1 - REVIEW

REVIEWER	Christiane Jannes 1. Research Unit Ethics, Institute for the History of Medicine and Medical Ethics, Faculty of Medicine, University of Cologne and University Hospital of Cologne, Cologne, Germany 2. Cologne Center for Ethics, Rights, Economics, and Social Sciences of Health (CERES), University of Cologne, Cologne, Germany
REVIEW RETURNED	20-Feb-2019

GENERAL COMMENTS	This is a relevant study for the area of family satisfaction with ICUs, with a very good design, large sample size and interesting achievements. However, there are a few things to consider before publication. Formal remark Please write the text in justification. Abstract The abstract is clear and well-written. Page 2, line 8: There is one space character too much. Page 2, line 20: A dot is missing at end of the sentence. Page 2, line 30: Please use the same spelling for the 1000s (with comma or without). Methods Page 5, line 24 ff.: When describing the recruitment process and the follow-up, it is not clear which data protection measures have been taken. How was the anonymity of the study participants guaranteed? Was there a blinding process etc.? Discussion Page 13: The result that relatives of deceased patients were more satisfied is an interesting and also not at first sight obvious result. A justification for this result is missing. What presumptions/statements
---

	exist for the result that these relatives were more satisfied? Also why the characteristics of the family members or patients can influence the satisfaction is not justified. The overall results should not only be discussed through the presentation of comparable study results, but also the background and justification should be emphasized. Overall, the entire results should be discussed in more detail. Furthermore, there are very few limitations presented, please specify. Tables Which threshold for significance is chosen and why? (p-value < 0,05?) For clarity, it would be advantageous if significant results in the tables were marked, for example, with an asterisk. References A uniform spelling is requested. Please write all surnames out.
--	---

REVIEWER	Nitin Puri Cooper University Hospital Camden, New Jersey, USA
REVIEW RETURNED	01-Mar-2019

GENERAL COMMENTS	The paper is excellent and I enjoyed reviewing it. I have few comments below. A. 94% of the patients were Caucasian, but UK ethnic make-up is only 86% Caucasian, can the discrepancy be explained and acknowledged in the manuscript. Also, in table 2, were white relatives of families more satisfied than other ethnic groups, it is unclear from the table. B. A similar point, that only 3% of the families did not have English as their first language, this seems small, but perhaps this is reflective of the UK's population. C. Table 2 -> What is neurological support? Is than an external ventricular drain or bolt? I believe clarifying this would be helpful. D. I would be interested in the author's opinion as to why family of non-survivors were happier with ICU care than families of survivors? E. Is a nursing ratio mandated in the UK for ICU patients? If it is, acknowledging this would be important, the manuscript seems to suggest it is not. F. What does NVQ stand for?
--

REVIEWER	Jeff Pan the Ohio State University, US
REVIEW RETURNED	13-Mar-2019

GENERAL COMMENTS	There are mistakes in table 2, specifically, page 8 (line 54--page 9 line 6) the Apache scale, also page 9 line 22, 25-31, 37. The statistical model did not consider the cap effect, or the skewed distribution of the satisfactory score. should be considered as a limitation at least. Page 11 line 5, There is no statistical evidence that the association is "independent". Please add the evidence or revise this claim; Table 4, the p values for the constant does not make any sense here.
---

VERSION 1 – AUTHOR RESPONSE

Reviewer: 1

Reviewer Name: Christiane Jannes

Institution and Country: 1. Research Unit Ethics, Institute for the History of Medicine and Medical Ethics, Faculty of Medicine, University of Cologne and University Hospital of Cologne, Cologne, Germany 2. Cologne Center for Ethics, Rights, Economics, and Social Sciences of Health (CERES), University of Cologne, Cologne, Germany

Please state any competing interests or state 'None declared': None declared

This is a relevant study for the area of family satisfaction with ICUs, with a very good design, large sample size and interesting achievements.

We thank the reviewer for these comments.

Formal remark

Please write the text in justification.

All text has been justified.

Abstract

The abstract is clear and well-written.

Page 2, line 8: There is one space character too much.

Page 2, line 20: A dot is missing at end of the sentence.

Page 2, line 30: Please use the same spelling for the 1000s (with comma or without). All points raised have been addressed in the Abstract.

Methods

Page 5, line 24 ff.: When describing the recruitment process and the follow-up, it is not clear which data protection measures have been taken. How was the anonymity of the study participants guaranteed? Was there a blinding process etc.?

All data were entered centrally onto a secure database. At the point of data entry all identifiable information was removed. Reference to this has been added into the Methods.

Discussion

Page 13: The result that relatives of deceased patients were more satisfied is an interesting and also not at first sight obvious result. A justification for this result is missing. What presumptions/statements exist for the result that these relatives were more satisfied?

The observational design of the study precludes any causative inferences being made. We could speculate that the higher levels of family satisfaction in the family members of nonsurvivors is due to: a) greater involvement of the family in end- of-life decision making; b) family members of survivors having ongoing issues to cope with following their family member's discharge from ICU; or c) other unknown reasons etc. To fully identify and understand the reasons, a detailed qualitative study would be required. We are not sure that we should speculate or make presumptions in the Discussion.

Also why the characteristics of the family members or patients can influence the satisfaction is not justified. The overall results should not only be discussed through the presentation of comparable study results, but also the background and justification should be emphasized. Overall, the entire results should be discussed in more detail.

Again, we can only speculate why characteristics of the family members or patients can influence family satisfaction. For instance, one possible hypothesis is that children, rather than parents of patients, may be more dissatisfied due to generational issues. One other hypothesis could be differences in expectations. We refer you the above response regarding speculation around the results in the Discussion.

Furthermore, there are very few limitations presented, please specify. We have now expanded on the study limitations in the Discussion.

Tables

Which threshold for significance is chosen and why? (p-value < 0,05?)

For clarity, it would be advantageous if significant results in the tables were marked, for example, with an asterisk.

The threshold for significance was set at $p < .05$. In line with The American Statistical Associations statement on P-values, we have supplemented the data summaries and pvalues with estimates of the effect and measures of their uncertainty (95% confidence intervals). We do not believe that dichotomising results into binary significant/nonsignificant using asterisks is necessary or the correct way to present the results.

References

A uniform spelling is requested. Please write all surnames out.

In line with BMJ Open guidelines for authors, references with more than three authors, the names of the first three authors are listed followed by 'et al.'

Reviewer: 2

Reviewer Name: Nitin Puri

Institution and Country: Cooper University Hospital, Camden, New Jersey, USA Please state any competing interests or state 'None declared': None

The paper is excellent and I enjoyed reviewing it. I have few comments below. We thank the reviewer for this comment.

A. 94% of the patients were Caucasian, but UK ethnic make-up is only 86% Caucasian, can the discrepancy be explained and acknowledged in the manuscript. Also, in table 2, were white relatives of families more satisfied than other ethnic groups, it is unclear from the table.

We have added to the higher proportion of Caucasian patients as a limitation in the Discussion. White family members were more satisfied than other ethnic groups. A statement addressing this has been added into the Discussion.

B. A similar point, that only 3% of the families did not have English as their first language, this seems small, but perhaps this is reflective of the UK's population.

We speculate that this is likely to reflect that 94% of patients were of white ethnicity, where only 1% reported that their first language was not English. Reference to this has been included above mentioned statement in the Discussion.

C. Table 2 -> What is neurological support? Is that an external ventricular drain or bolt? I believe clarifying this would be helpful.

Neurological support (as collected in the Case Mix Programme national clinical audit) is defined as: admissions with central nervous system depression sufficient to prejudice their airway and protective reflexes; admissions receiving invasive neurological monitoring or treatment (e.g. ICP (intracranial pressure), jugular bulb sampling, external ventricular drain etc.); admissions receiving continuous intravenous medication to control seizures and/or for continuous cerebral monitoring; and admissions receiving therapeutic hypothermia using cooling protocols or devices. An abridged definition is now provided in the legend of Table 2.

D. I would be interested in the author's opinion as to why family of non-survivors were happier with ICU care than families of survivors?

Please see earlier response on to comment 4 from Reviewer 1.

E. Is a nursing ratio mandated in the UK for ICU patients? If it is, acknowledging this would be important, the manuscript seems to suggest it is not.

Nurse staffing ratios are mandated in the UK. Nurse/patient ratios of 1:1 and 1:2 are mandated for Level 3 (Intensive Care) and Level 2 (High Dependency) patients, respectively. Reference to this has been added into the Methods.

F. What does NVQ stand for?

National Vocational Qualifications (NVQ) are a competency qualification in England, Wales and Northern Ireland that recognise skills or knowledge specific to the type of employment. NVQ equivalents were used to identify the highest level of education where NVQ level 1 or 2 is equivalent to GCSE or O-level (school exams taken at age 16); NVQ level 3 is equivalent to A-level, AS-level or High School Certificate (school exams taken at age 18); NVQ level 4 or 5 is equivalent to degree, Higher degree, Higher National Certificate, Higher National Diploma. This definition has been added in the legend to Table 1.

Reviewer: 3

Reviewer Name: Jeff Pan

Institution and Country: the Ohio State University, US Please state any competing interests or state 'None declared': None declared

There are mistakes in table 2, specifically, page 8 (line 54--page 9 line 6) the Apache scale, also page 9 line 22, 25-31, 37. These have now been corrected.

The statistical model did not consider the cap effect, or the skewed distribution of the satisfactory score. should be considered as a limitation at least.

When assessing patient and family satisfaction, it is not uncommon for continuous measures to be skewed. Skewness is not a problem by itself, but it might lead to the violation of normality assumptions in multilevel analyses. In general, failure of the normality assumption at response level or group level will not bias estimation of the fixed effects (Gelman & Hill 2007; Maas & Hox 2004), but it might cause problems when one is interested in the significance or in the confidence intervals of the variance terms at the group level (Maas & Hox 2004). One method to obtain better tests and confidence intervals is to correct the asymptotic standard errors. We used a robust (Huber/White) variance estimator to calculate robust standard errors.

After modelling, the normality of error assumption was assessed by measurements of skewness. Normal probability plots were also used to assess the distribution of residuals at each level. In the model for overall family satisfaction among family members of ICU survivors, the skewness statistics for level 1, 2 and 3 scores were -0.79, -0.96 and -0.50, respectively. For family members of ICU non-survivors, the corresponding skewness statistics were -1.26, -0.83 and -0.21. The normal probability plots for standardized residuals at level-3 (unit) were close to linearity for both models, although the lower tail was a bit heavy in the plots at the patient level and response level.

One method of dealing with violation of the normality assumption is to transform the outcome variable to improve the error distribution. Therefore, as a sensitivity analysis we ran a multilevel regression model on the square root of the score using the same set of variables to assess inference. Results were consistent with the non-transformed score.

We have added reference to the sensitivity analyses into the Methods and Results sections. Although skewness of the scores did not lead to concerns in the present work, the skewness of satisfaction scores is considered a study limitation and has been added to the Discussion.

Page 11 line 5, There is no statistical evidence that the association is "independent".

Please add the evidence or revise this claim;

By "independently associated", we meant that a variable had a significantly non-zero coefficient in an adjusted multilevel model. We have changed this wording to state that the predictors "were significant when controlling for other predictors in the model".

Table 4, the p values for the constant does not make any sense here. The p values for the constant have been removed from Table 4.

VERSION 2 – REVIEW

REVIEWER	Christiane Jannes 1. Research Unit Ethics, Institute for the History of Medicine and Medical Ethics, Faculty of Medicine, University of Cologne and University Hospital of Cologne, Cologne, Germany 2. Cologne Center for Ethics, Rights, Economics, and Social Sciences of Health (CERES), University of Cologne, Cologne, Germany
REVIEW RETURNED	07-Jun-2019

GENERAL COMMENTS	The paper has gained a lot from the revision. I like the preparation very much. Just one important remark I would ask you to revise. Since two of the reviewers have noted the point, I would consider it important to integrate the interpretation or own explanation of the point that "relatives of deceased patients are more satisfied" into the discussion. Exactly as you noted in your statement, you could point out the possible explanations and recommend a qualitative methodology for answering this question. I wish you continued success!
--

REVIEWER	Nitin Puri Cooper Medical School of Rowan Univeristy
REVIEW RETURNED	05-Jun-2019

GENERAL COMMENTS	The authors have addressed my concerns.
---

VERSION 2 – AUTHOR RESPONSE

Reviewer: 1

Reviewer Name: Christiane Jannes

Institution and Country: 1. Research Unit Ethics, Institute for the History of Medicine and Medical Ethics, Faculty of Medicine, University of Cologne and University Hospital of Cologne, Cologne, Germany

2. Cologne Center for Ethics, Rights, Economics, and Social Sciences of Health (CERES), University of Cologne, Cologne, Germany

Please state any competing interests or state 'None declared': None declared

Please leave your comments for the authors below

The paper has gained a lot from the revision. I like the preparation very much.

Just one important remark I would ask you to revise. Since two of the reviewers have noted the point, I would consider it important to integrate the interpretation or own explanation of the point that "relatives of deceased patients are more satisfied" into the discussion. Exactly as you noted in your statement, you could point out the possible explanations and recommend a qualitative methodology for answering this question.

I wish you continued success!

As requested, we have added our opinions on why family members of non-survivors might be more satisfied than family members of survivors into the Discussion.

Reviewer: 2

Reviewer Name: Nitin Puri

Institution and Country: Cooper Medical School of Rowan Univeristy

Please state any competing interests or state 'None declared': None

Please leave your comments for the authors below

The authors have addressed my concerns.

No response required.